# Partial Convolution Network for Metal Artifact Reduction in CT Preprocessing: Preliminary Results

**Laura Hellwege**                hellwege@imt.uni-luebeck.de
**Nele Blum**                  blum@imt.uni-luebeck.de
*University of Lübeck - Institute of Medical Engineering, Ratzeburger Allee 160,*
*23562 Lübeck, Germany*

**Thorsten M. Buzug**           thorsten.buzug@imte.fraunhofer.de
**Maik Stille**               maik.stille@imte.fraunhofer.de
*Fraunhofer IMTE, Mönkhofer Weg 239a, 23562 Lübeck, Germany*

**Editors:** Under Review for MIDL 2021

## Abstract

Metal artifacts impair the diagnostic value of medical CT images. These artifacts occur from the projection values associated with the metal objects inside the scanned anatomy. In this work, we replace the corrupted projection values by using a deep convolutional neural network consisting of so-called partial convolution layers. We show that the network trained on simulated data enhances newly presented projection data and therefore the corresponding reconstructed image.

**Keywords:** Metal artifact reduction, CT preprocessing, projection domain, CNN, partial convolution.

## 1. Introduction

The aim of this paper is to reduce metal artifacts in cone-beam computed tomography using a deep convolutional neural network (CNN). The network operates on CT projection data by replacing impaired projection values using the information of the spatially surrounding data. This task is also known as inpainting. In the field of CT preprocessing, meaning the enhancement of CT projection data exclusively before reconstruction, three different approaches by (Park et al., 2017), (Gjesteby et al., 2017), and (Liao et al., 2019) exist to our knowledge. Outside the field of CT preprocessing, one particularly interesting approach to image inpainting is presented by (Liu et al., 2018). Here, the convolution results are adapted using additional binary masks in every layer of the network. In the following, this idea is adapted to CT preprocessing.

## 2. Methods

We generate training data by performing a cone-beam forward projection of 90 human phantom datasets with different anatomies. Multiple small metal objects are randomly added to the anatomies before projecting. Afterwards, the metal trace is deleted based on a forward-projected segmentation of the metal object. The corresponding ground-truth data consists of the projection values of the same anatomy without metal insertion. The 3D projection data is standardized and randomly sampled into 2D sinogram patches of size

Table 1: Properties of encoding and decoding PC (EPC/DPC) layers.

| Module Name | Input Size | Kernel Size | # Channels | Stride Factor | Activation |
|---|---|---|---|---|---|
| EPC($S$, $K$, $C$) | $S \times S$ | $K \times K$ | $C$ | 2 | ReLU |
| DPC($S$, $K$, $C$) | $S \times S$ | $K \times K$ | $C$ | 1 | LeakyReLU(0.2) |

$128 \times 128$. As second input to the network, an equally-sized binary mask of the metal trace is used. For inpainting enhancement, (Liu et al., 2018) introduced the so-called partial convolutional (PC) layer. In a PC layer the value $x'$ resulting from a convolution of the previous feature map $X$ windowed by kernel $W$ is defined as

$$x' = \begin{cases} W^T(X \odot M)r(M), & \text{if } \|M\|_1 > 0 \\ 0, & \text{otherwise} \end{cases}$$

where $r(M) = \frac{\text{size}(M)}{\|M\|_1}$ is a correction factor depending on the windowed binary mask $M$. $\odot$ denotes a element-wise multiplication. At the positions for which $x'$ is non-zero, the mask $M$ is updated by setting its value to one. Thus, the number of invalid values is reduced layer by layer. The implementation of the PC layer and network are adapted from (Gruber, 2019). The network has a U-net structure which layer properties are shown in Table 1. The encoding consists of the layers EPC1(128, 3, 64) without batch normalization followed by EPC2(64, 3, 128), EPC3(32, 3, 256), EPC4(16, 3, 512), EPC5(8, 3, 512) and EPC6(4, 3, 512) with batch normalization. The decoding is performed by nearest-neigbor upsampling, concatenation with the same-sized encoding result, and PC. The PC layers are DPC1(8, 3, 512), DPC2(16, 3, 256), DPC3(32, 3, 128) and DPC4(64, 3, 64). The last layer DPC5(128, 3, 1) has no activation function. As suggested by (Liu et al., 2018), the training of the network is performed with the ADAM optimizer of learning rate 0.0002 with a batch size of six. The loss function consists of the sum of the mean absolute error (MAE) evaluated on the masked region and a discrete total variation term on the masked region's border.

## 3. Results and Discussion

Table 2: Mean values of MSE, MAE and SSIM computed on 15 test anatomies. The first value in each cell corresponds to evaluation in the projection domain, the second value to evaluation on the reconstructed image.

| Method | MSE | MAE | SSIM |
|---|---|---|---|
| BLI | 4.4E-3 / 4.61 | 3.8E-2 / 1.23 | 0.993 / 0.997 |
| CNN | 1.1E-2 / 137.13 | 6.4E-2 / 6.13 | 0.987 / 0.986 |
| PCNN | 9.8E-3 / 81.50 | 6.6E-2 / 5.22 | 0.987 / 0.990 |

In the projection domain, the inpainting results are evaluated by computing mean squared error (MSE), mean absolute error (MAE) and structural similarity (SSIM). In the image

domain, a single value is calculated for one complete 3D volume outside the metal object. The results for all metrics can be found in Table 2. For comparison, bilinear interpolation (BLI) is performed in the projection domain. Additionally, the performance of a CNN with regular convolution layers and the same architecture as the PCNN is evaluated. For all metrics, we obtain similar results which indicate that the PCNN prediction improves the image quality of the reconstructed image. However, it does not show superiority to bilinear interpolation. Figure 1 shows a visual comparison between the three correction methods. Our results suggest that the PC layer alone is not the reason for the excellent network performance presented by (Liu et al., 2018). A different loss function and more diverse training data might be needed for further improvement. As for the network design, the results could be improved by using the full 2D sinogram as input to reduce inconsistencies. However, the differences of errors in projection and image domain of PCNN and CNN already indicate higher consistency of the PCNN method. Additionally, a different correction term and mask update for the PC layer could enhance the accuracy of the corrupted convolution results.

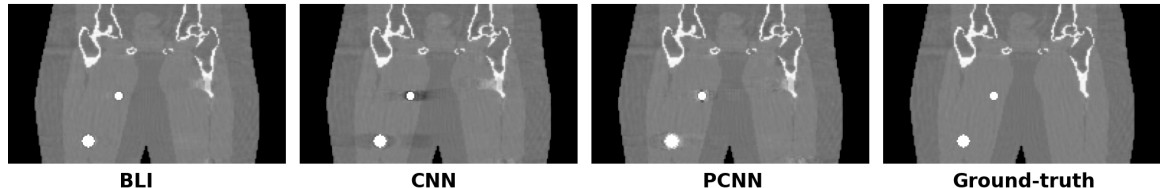

Figure 1: Examplary reconstruction results after correction in the projection domain. The images are displayed in HU (level: 300, window: 500).

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
