# OpenReview forum: "Partial Convolution Network for Metal Artifact Reduction in CT Preprocessing: Preliminary Results"
_MIDL.io/2021/Conference/Short — MIDL 2021 Poster_

### Official Review · Reviewer_ADzG · 2021-04-25

**Confidence:** 4
**Final Rating:** 3

**Summary:**

The authors use partial convolutional layers for CT inpainting to remove artifacts by metals. The proposed method is compared to bilinear interpolation and classical convolutions showing that partial convolutions are improving the image quality but is only on par with bilinear interpolation. The authors suggest several potential improvements to their methodology, such as a different correction term and a more diverse dataset, to improve the performance.

**Strengths:**

The removal of metal artifacts are an important problem to be addressed in CT reconstruction. The use of partial convolutions as a potentially very powerful inpainting technique is therefore of high interest. The fair comparison with classical convolutions and a simple bilinear interpolation allows the accurate judgement of the presented method. The paper is also clearly written, the motivation is straight forward and in general easy to follow. The authors also provide appropriate code to reproduce the results.

**Weaknesses:**

•	I find the description of the training data generation is a little bit weak. What data is exactly used? Is it from a public dataset? How large were the metal objects, how many were inserted per phantom, were they placed in physiological positions?
•	For Figure 1, the reviewer would like to have seen the corrupted image, the mask and the image with holes to be reconstructed to judge the complexity of the task


**Deanonymize Review:**

no

**Justification Of The Rating:**

Even though that the proposed method is “only” the application of a previously proposed method onto different data, I think the fact that partial convolutions should be preferably used compared to pure CNNs and the fair comparison to bilinear interpolation is potentially interesting to MIDL.

**Paper Type:**

validation/application paper

**Special Issue:**

no

---

### Official Review · Reviewer_NY3c · 2021-04-30

**Confidence:** 3
**Final Rating:** 3

**Summary:**

This paper validated the effectiveness of partial convolution network for inpainting problem in CT to reduce metal artifact. The results showed non-significant improvement of PCNN over baseline U-Net. Besides, deep learning based results were worse than bilinear interpolation. Future work is also discussed in the paper.

**Strengths:**

1. The motivation of this work was clear and meaningful, as impaired CT projection data recovery helps improve image quality.
2. The data collection process was rigorous which generated unique training data for the task.

**Weaknesses:**

It's concerning that U-Net and PCNN were worse that bilinear interpolation, as nowadays from literature CNN-based methods usually outperform iterative methods in medical imaging tasks. Obviously further work should be done such as hyperparameter optimization (batchsize, stepsize, network architecture, etc) and generating more training data.

**Deanonymize Review:**

no

**Justification Of The Rating:**

Data collection with domain knowledge makes this paper rigorous. Comparison between conventional methods and CNNs implied that further work such as hyper-parameter optimization and data generation is needed to improve CNN performance.

**Paper Type:**

validation/application paper

**Special Issue:**

no

---

### Meta-Review · Program_Chairs · 2021-05-09

**Recommendation:** Accept (Poster)
**Confidence:** 4

**Metareview:**

Reviewers are unanimous in their recommendation to accept this paper.

---

### Decision · Program_Chairs · 2021-05-11

Accept (Poster)